

# Diversity of rhizosphere and endophytic fungi in *Atractylodes macrocephala* during continuous cropping

Bo Zhu[1],[*], Jianjun Wu[1],[*], Qingyong Ji[2], Wei Wu[1], Shihui Dong[1], Jiayan Yu[1], Qiaoyan Zhang[1] and Luping Qin[1]

[1] School of Pharmacy, Zhejiang Chinese Medical University, Hangzhou, China
[2] Lishui Academy of Agricultural and Forestry Sciences, Lishui, China
[*] These authors contributed equally to this work.

## ABSTRACT

Rhizospheric and endophytic fungi are key factors which influence plant fitness and soil fertility. *Atractylodes macrocephala* is one of the best-known perennial herbs used in traditional Chinese medicine. Continuous cropping has been shown to have a negative effect on its growth and renders it more susceptible to microbial pathogen attacks. In this study, we investigated the effects of continuous cropping on the endophytic and rhizospheric fungi associated with *A. macrocephala* using culture-independent Illumina MiSeq. Continuous cropping was found to decrease fungal diversity inside plant roots, stems, leaves and tubers. Additionally, we found that the structure and diversity of rhizospheric and endophytic fungal communities were altered by root-rot disease. *Fusarium* was overrepresented among root-rot rhizospheric and endophytic fungi, indicating that it has a major negative impact on plant health during *A. macrocephala* monocropping. Canonical correspondence analysis of the control and diseased samples revealed that pH, hydrolysis N, electrical conductivity and Hg content were well-correlated with fungal community composition during continuous cropping. Taken together, these results highlight the ecological significance of fungal communities in maintaining plant fitness and will guide the development strategies to attenuate the negative impacts of *A. macrocephala* continuous cropping.

## INTRODUCTION

Plants and rhizospheric soil are colonized by fungal communities that can impact their fitness by influencing nutrient acquisition, causing soil-borne diseases and affecting the activity of plant pathogens (*Tan et al., 2017a*). Many previous studies have shown that studying plant-microorganism interactions can lead to improvements in several agronomic processes, including crop rotation and tillage (*Somenahally et al., 2018*), pesticide application (*Regar et al., 2019*), irrigation (*Dang et al., 2019*), fertilizer application (*Nguyen et al., 2018*) and continuous cropping (*Ali et al., 2019*; *Xiong et al., 2015*). Continuous cropping is widely adopted in Chinese agricultural production and is defined as cultivating

Corresponding authors
Qiaoyan Zhang, zqy1965@163.com
Luping Qin, lpqin@zcmu.edu.cn

the same or similar crop species for a long period of time (*Shipton, 1977*). Studies have shown that continuous cropping leads to compromised growth, yield loss, disease susceptibility and quality deterioration (*Hontoria et al., 2019*). The detrimental effects of continuous cropping have been demonstrated in a variety of crop species, including *Atractylodes macrocephala* (*Zheng et al., 2018*).

*A. macrocephala* is a perennial herb that has been cultivated for over 700 years in temperate and subtropical regions. The tuber of *A. macrocephala* is commonly referred to as "*Baizhu*", and has been used to treat cancer, osteoporosis, gastrointestinal dysfunction, obesity and fetal irritability in traditional Chinese medicine in East Asia (*Zhu et al., 2018b*). It has recently been shown that continuous cropping of *A. macrocephala* can lead to reduced yield and quality (*Zheng et al., 2018*), possibly due to alterations in soil enzymatic activities, allelochemical substance enrichment, soil microbial community changes or soil-borne pathogen accumulation (*Xiong et al., 2015*). Change in soil microbiota communities has been singled out as one of the major causes of yield loss during *A. macrocephala* continuous cropping (*Chen et al., 2014*; *Shi, 2018*). Root-rot diseases, which are often associated with *A. macrocephala* continuous cropping, also play a major role in loss of yield and quality during *A. macrocephala* continuous cropping (*Zheng et al., 2018*). Most of these diseases are associated with fungal pathogens such as *Fusarium oxysporum*, *Rhizoctonia solani*, and *Ceratobasidium* sp. (*Liu, 2012*; *You et al., 2013*; *Zhang, 2015*).

Rhizospheric fungi play key roles in organic matter decomposition, nutrient cycling and soil fertility maintenance (*Miao et al., 2016*; *Zhou et al., 2017*). Some rhizospheric fungi have also been associated with pathogen growth inhibition, leading to their application as biocontrol agents (*Venneman et al., 2019*). Additionally, the overall level of endophytic fungi has been shown to be a good indicator of host plant health (*Zhu et al., 2018a*). Recent studies have proposed that continuous cropping results in imbalances in endophytic and rhizospheric soil fungal community diversity and structure, and rapid accumulation of fungal pathogens (*Liu et al., 2019*; *Qin et al., 2017*). Thus, a healthy and stable endophytic and rhizospheric fungal community may be essential for maintaining long-term continuous cropping and stable crop yields.

Both the mechanism by which continuous cropping affects rhizospheric soil fungal communities and how these changes influence soil productivity are still largely unknown. To date, there has been no culture-independent studies of rhizospheric soil and endophytic fungal community diversity during monocropping of *A. macrocephala* (*Xiong et al., 2015*; *Zhou & Wu, 2012*). In this study, we used molecular characterization to examine changes in rhizospheric fungal communities, rotation soils, roots, leaves, stems, and tubers in *A. macrocephala* grown in continuous cropping fields. We aimed to investigate the link between soil physiochemical properties and the structure, composition and diversity of fungal communities. Additionally, we examined the underlying mechanism by which continuous cropping of *A. macrocephala* influenced endophytic and rhizospheric fungal communities.

## MATERIALS & METHODS

### Study site and experimental design

Samples were collected from a grassland located in a hilly area in Lin'an County, Zhejiang Province, China, with a typical mid-subtropical monsoon climate (30°14′N, 119°25′E, 155 m elevation). The study area was divided into five zones according to different cropping practices (no *A. macrocephala* planted, *A. macrocephala* planted for 1 year, *A. macrocephala* planted for 2 years, 1-year fallow, and 2-year fallow). Each zone consisted of a $20 \times 20$ m plot, which was managed as uniformly as possible. Due to limited land access, only one plot was used for each zone. However, we randomly collected 15 soil cores from the plow layer (0–30 cm in depth), thoroughly pooled them as a composite sample and divided the pooled samples into three replicates to make the samples more representative (*Hu et al., 2017*; *Tan et al., 2017a*). Fallow soil zones were left unplanted for the specified number of years after an initial planting and harvesting of *A. macrocephala*. For the fallow zones, three soil samples were randomly selected, with each soil sample composed of 15 soil cores randomly collected from the plow layer (0–30 cm in depth).

Plants were sampled by inspecting them for root rot symptoms, and were then placed either into the group with or without symptoms. Symptoms were recorded regarding the general appearance of the entire plant, and more specifically regarding the health of the roots, the stems, and the foliage. In the absence of overt symptoms in above-ground parts, roots were checked for necrotic flecking, dead, broken roots, and water-soaked lesions (*Sims et al., 2019*). For the 1-year and 2-year *A. macrocephala* planted zones, 15 plant samples for each plant type (1-year healthy plant, 2-year healthy plant, and 2-year diseased plant) were collected and put into sterile plastic bags, placed on ice and transported to the laboratory immediately. The soil which was loosely attached to the roots was removed by gentle shaking. Tightly bound soil was considered rhizospheric and was obtained by firmly shaking the root system in a sterile plastic bag (*Kobayashi et al., 2015*; *Tan et al., 2017b*). The rhizospheric soil samples were thoroughly mixed, and randomly divided into three subsamples, as described previously (*Yu et al., 2019*). The plants were washed with tap water and then rinsed three times with distilled water. They were then separated into leaf, stem, tuber, and root tissues. For surface disinfection, samples from each plant tissue were successively immersed in 75% ethanol for 1.5 min, a fresh 2.5% sodium hypochlorite solution for 3 min, and 75% ethanol for 30 s, then washed with distilled water three times (*Zhu et al., 2018a*). Each plant tissue was thoroughly mixed to make a composite sample, and randomly divided into three subsamples. In total, there were 18 soil samples (3 fallow soils and 3 rhizospheric soils, with 3 replicates each) and 36 plant samples (3 plant types, with 4 tissue types and 3 replicates each). Collected soils were sieved through a <4 mm filter and stored at 4 °C prior to biochemical characterization, or frozen at −80 °C less than two weeks before DNA extraction.

The soil and plant conditions were as follows: 1-year fallow soil (FS_1), 2-year fallow soil (FS_2), 1-year healthy rhizospheric soil (RS_H1), 2-year healthy rhizospheric soil (RS_H2), 2-year root-rot diseased rhizospheric soil (RS_D), blank control soil with no *A. macrocephala* planted (S_CK), 1-year-old healthy root/stem/leaf/tuber (R/S/L/T_H1),

2-year-old healthy root/stem/leaf/tuber (R/S/L/T_H2), 2-year-old root-rot diseased root/stem/leaf/tuber (R/S/L/T_D) (Fig. S1).

## Soil physicochemical properties

The chemical and physical properties of all soil samples were analyzed. Soil pH was measured by a pH meter and soil electrical conductivity by an electric conductivity bridge, according to the manufacturers' protocols (*Satapute et al., 2019*). The moisture content of the soil was determined after dry weights were obtained by drying the samples at 105 °C. The soil organic matter content was measured using Yang's method (*Yang et al., 2019*). We used a nickel crucible to digest soil total K with sodium hydroxide at 750 °C, and extracted soil available K using 1 mol/L ammonium acetate (*Zhao et al., 2014*). All K contents were determined by an atomic absorption spectrophotometer (AAnalyst 400, PerkinElmer, US) (*Yao et al., 2017*). Soil available P and total P and were determined by the ammonium molybdate method (*Murphy & Riley, 1962*). Soil total N was determined using the semi-micro Kjeldahl method, according to Nelson and Sommers (*Nelson & Sommers, 1980*). Soil hydrolysis N was determined by the alkali-hydrolyzed distillation method described by *Zhang et al. (2018)*. Heavy metals Pb, Cu, Hg, As, Cr and Cd were detected using flame atomic absorption spectrometry (Analytikjena AAS vario6, Germany) according to the method of *Ye et al. (2011)*. All assays were performed in triplicate.

## DNA extraction, PCR amplification, and sequencing

Genomic DNA was isolated from the rhizosphere, fallow soil, blank control soil samples and plant organs using the E.Z.N.A.® soil DNA Kit (Omega Bio-tek, USA) according to the manufacturer's instructions. The DNA extractions for each sample were conducted in triplicate. The quantity and purity of the DNA samples were assayed by a NanoDrop 2000 UV-vis spectrophotometer (Thermo Scientific, USA). The DNA samples were also run on a 1% agarose gel by electrophoresis to further assess their quality and integrity. The DNA samples were diluted to 1 ng/$\mu$L with sterile water and stored at $-80$ °C until further processing. The ITS1 region was amplified from fungal genomes with primers ITS1-F (CTTGGTCATTTAGAGGAAGTAA) and ITS2-F (GCTGCGTTCTTCATCGATGC) in a thermocycler PCR system (Eppendorf Mastercycler pro, Germany). The A adaptor-containing end was the sequencing end, and a 10 bp barcode was added between the adaptor and the forward primer sequence to allow for dis-crimination of the samples during sequencing (*Bai et al., 2015*). The PCR mixtures contain 5× *TransStart* FastPfu buffer 4 $\mu$L, 2.5 mM dNTPs 2 $\mu$L , forward primer (5 $\mu$M) 0.8 $\mu$L, reverse primer (5 $\mu$M) 0.8 $\mu$L, *TransStart* FastPfu DNA Polymerase 0.4 $\mu$L, template DNA 10 ng, and finally ddH$_2$O up to 20 $\mu$L. The reaction program was: initial denaturation at 95 °C for 3 min, followed by 27 cycles of denaturing at 95 °C for 30 s, annealing at 55 °C for 30 s and extension at 72 ° C for 45 s, and single extension at 72 °C for 10 min, and end at 4 °C. The PCR reactions for each DNA sample were conducted in triplicate, resulting in nine total PCR products for each plant/soil sample (3 DNA replicates, with 3 PCR replicates each). The resulting amplicons were separated by electrophoresis using a 2% agarose gel and extracted with the AxyPrep DNA Gel Extraction Kit (Axygen Biosciences, USA). The

nine purified PCR products were then pooled before paired-end sequencing ($2 \times 300$) on an Illumina MiSeq platform (Illumina, USA) (*Saenz et al., 2019*).

## Bioinformaitics analysis

The raw ITS rDNA gene sequencing reads were demultiplexed, quality filtered by Trimmomatic and merged by FLASH with the following criteria: (1) Reads were removed if any site had an average quality score <20 over a 50 bp sliding window. Reads containing Ns or with a length of less than 50 were also removed. (2) The remaining pair-end reads were assembled into larger contigs according to their overlaps, with a minimum overlap length of 10 bp. (3) The maximum mismatch ratio allowed in the overlap area of the merged sequence was 0.2. (4) The directionality of reads was corrected based on their barcodes and primer sequences, with no mismatches allowed in the barcode and 2 mismatches allowed in the primers (FLASH and Trimmomatic).

We assigned sequences with ≥ 97% similarity to the same Operational Taxonomic Units (OTUs) in UPARSE (version 7.1, Table 1) and then selected a representative OTU for each community in order to perform taxonomic annotation using a Ribosomal Database Project classifier. The OTU clustering process and criteria were as follows: (1) In order to reduce the amount of redundant calculation in the process of analysis, non-repetitive sequences were obtained from optimized data (http://drive5.com/usearch/manual/dereplication.html). (2) Singletons were removed (http://drive5.com/usearch/manual/singletons.html). (3) With 97% clustering, an OTU sequence was required to be at least 3% different from all other OTUs, and OTU representative sequence were the most abundant sequences in their neighborhood. During the clustering process, chimeric sequences were discarded. (4) All optimized sequences were mapped to OTU representative sequences, and sequences with a similarity >97% were selected to generate an OTU table (Usearch, vsesion 7.0, http://drive5.com/uparse/). Alpha within samples was analyzed by in-house Perl scripts.

Next, we rarified the OTU table and calculated Observed Species, Shannon's index, and Chao1to calculate Alpha Diversity (*Tan et al., 2017a*). QIIME and the compute_core_microbiome.py script were used to identify OTUs that occurred within 95% of the *A. macrocephala* or soil samples (*Mahoney, Yin & Hulbert, 2017*). A species was defined as dominant if $P_i > 1/S$, where S represents species richness, and $P_i$ is the relative abundance of a species i, defined as the number of competing species present in the community (*Rivera-Orduna et al., 2011*; *Wu et al., 2013*).

## Statistical analysis

One-way ANOVA of soil physical and chemical properties as well as the changes in dominant fungal genera of rhizospheric and fallow soil samples were performed using SPSS (version 19.0; SPSS, Chicago, IL, USA). Significance was calculated by Duncan's mean test ($P < 0.01$). Mean values of replicates are expressed as mean ± standard error (SE). False discovery rate (FDR) was used to adjust the significance value in multiple comparisons.

A heatmap was drawn to show the relative abundances of the 25 predominant genera in each sample. Principle component analysis (PCA) was conducted using all OTUs relative abundances in the plant and soil samples.
**Table 1  Summary of data and alpha-diversity of the fungal community of rhizospheric soil, plant endophytic and fallow soil samples.** Each plant and soil samples (18 samples) were randomly divided into three subsamples (18 × 3), and were analyzed using the average data of their sub-samples, respectively. Different letters in one column denote significant differences at $P < 0.01$ among the samples from same source. One-year fallow soil (FS_1), 2-year fallow soil (FS_2), 1-year healthy rhizosphere soil (RS_H1), 2-year healthy rhizospheric soil (RS_H2), 2-year root-rot diseased rhizospheric soil (RS_D), blank control soil (no A. macrocephala was planted) (S_CK), 1-year-old healthy root/stem/leaf/tuber (R/S/L/T_H1), 2-year-old healthy root/stem/leaf/tuber (R/S/L/T_H2), 2-year-old root-rot diseased root/stem/leaf/tuber (R/S/L/T_D).

| Source of sample | Sample | Raw reads | Average length | Total bases ($\times 10^7$) | Q30 | Q20 | Nseqs | Coverage | OTU (97%) | OTU (95%) | Chao1 (97%) | Shannon (97%) |
|---|---|---|---|---|---|---|---|---|---|---|---|---|
| Soil | FS_1 | 58,691 | 265.37 | 1.56 | 97.87 | 99.31 | 33,919 | 0.9965 | 684 | 613 | 736.649C | 5.264C |
| | FS_2 | 42,763 | 270.10 | 1.16 | 97.66 | 99.24 | 33,919 | 0.9991 | 586 | 527 | 782.019B | 5.573B |
| | RS_H1 | 58,643 | 270.71 | 1.59 | 97.61 | 99.19 | 33,919 | 0.9961 | 614 | 498 | 645.101D | 4.213D |
| | RS_H2 | 51,435 | 269.26 | 1.38 | 97.77 | 99.28 | 33,919 | 0.9959 | 555 | 451 | 598.489E | 3.646F |
| | RS_D | 48,644 | 273.85 | 1.33 | 97.39 | 99.14 | 33,919 | 0.9968 | 500 | 411 | 545.750F | 3.722E |
| | S_CK | 40,379 | 272.72 | 1.10 | 97.59 | 99.21 | 33,919 | 0.9979 | 657 | 562 | 828.137A | 5.813A |
| Root | R_H1 | 59,575 | 263.53 | 1.57 | 98.49 | 99.51 | 33,919 | 0.9983 | 252 | 209 | 255.724B | 2.089C |
| | R_H2 | 34,464 | 257.25 | 0.89 | 98.50 | 99.41 | 33,919 | 0.9999 | 132 | 119 | 313.500A | 3.359A |
| | R_D | 44,330 | 268.88 | 1.19 | 98.11 | 99.40 | 33,919 | 0.9988 | 205 | 176 | 218.941C | 2.612B |
| Tuber | T_H1 | 48,878 | 256.17 | 1.25 | 98.64 | 99.56 | 33,919 | 0.9994 | 153 | 132 | 154.353A | 1.867B |
| | T_H2 | 48,295 | 265.07 | 1.28 | 98.21 | 99.41 | 33,919 | 0.9997 | 168 | 134 | 151.000B | 1.905A |
| | T_D | 57,338 | 244.44 | 1.40 | 98.72 | 99.57 | 33,919 | 0.9998 | 23 | 15 | 26.200C | 0.314C |
| Stem | S_H1 | 58,464 | 249.31 | 1.46 | 98.67 | 99.58 | 33,919 | 0.9979 | 345 | 275 | 350.143B | 3.109A |
| | S_H2 | 66,439 | 245.48 | 1.63 | 98.60 | 99.55 | 33,919 | 0.9971 | 321 | 240 | 393.500A | 2.519B |
| | S_D | 55,367 | 244.13 | 1.35 | 98.70 | 99.55 | 33,919 | 0.9998 | 106 | 91 | 106.750C | 1.211C |
| Leaf | L_H1 | 62,977 | 252.25 | 1.59 | 98.54 | 99.53 | 33,919 | 0.9975 | 332 | 247 | 337.893A | 2.942A |
| | L_H2 | 54,317 | 251.72 | 1.37 | 98.35 | 99.45 | 33,919 | 0.9980 | 258 | 202 | 322.500B | 2.348B |
| | L_D | 46,142 | 257.35 | 1.19 | 98.63 | 99.55 | 33,919 | 0.9990 | 250 | 205 | 251.667C | 2.191C |

We performed a canonical correspondence analysis (CCA) using normalized OTU abundance and soil physicochemical data using the vegan package implemented in R (https://www.r-project.org/) (*Dixon, 2003*). Significant correlations between soil properties and the fungal communities at OTU level were determined via Spearman's correlations using >0.8 or <−0.8 as the threshold, with a cutoff of $P <0.01$ (*Faust et al., 2012*). All the data analysis for correlations was finished in online 'i-sanger' (http://www.i-sanger.com/) developed by Majorbio Bio-Pharm Technology Co. Ltd (*Lin et al., 2018*).

## RESULTS

### Soil properties

Continuous cropping practice noticeably decreased soil pH compared with the S_CK and fallow soil. Electrical conductivity, total P, total K, hydrolysis N, available P, and available K contents were much higher in the continuous cropping soil relative to the S_CK, while organic matter and total N contents were higher in the S_CK than in continuous cropping soil samples. In addition, the continuous cropping soil exhibited much higher Pb, As, Cd, Cr, and Cu contents, while S_CK had higher Hg levels (Table 2).

Zhu et al. (2020), *PeerJ*, DOI 10.7717/peerj.8905

**Table 2 Physical and chemical nature of rhizospheric and fallow soils in Lin'an, China ($n = 3$).** Different letters in one column denote significant differences at $P < 0.01$, 1-year fallow soil (FS_1), 2-year fallow soil (FS_2), 1-year healthy rhizosphere soil (RS_H1), 2-year healthy rhizospheric soil (RS_H2), 2-year root-rot diseased rhizospheric soil (RS_D), blank control soil (no A. macrocephala was planted) (S_CK).

| Sample | pH | Organic matter (g/kg) | Total N(g/kg) | Total P(g/kg) | Total K(g/kg) | Hydrolysis N(mg/g) | Available P(mg/g) | Available K(mg/g) | Electrical conductivity (mS/m) | Pb(mg/g) | As(mg/g) | Hg(mg/g) | Cd(mg/g) | Cr(mg/g) | Cu(mg/g) |
|---|---|---|---|---|---|---|---|---|---|---|---|---|---|---|---|
| S_CK | 5.84 ± 0.16A | 44.0 ± 2.8AB | 1.73 ± 0.16AB | 0.24 ± 0.04B | 15.2 ± 0.7B | 163.2 ± 6.9A | 2.6 ± 0.4E | 68 ± 7D | 3.9 ± 0.3C | 18.6 ± 0.5BC | 2.84 ± 0.20C | 0.084 ± 0.002AB | 0.12 ± 0.01BC | 14 ± 2C | 22 ± 2A |
| RS_H1 | 5.55 ± 0.25AB | 41.4 ± 1.4BC | 1.60 ± 0.21AB | 0.30 ± 0.05AB | 16.8 ± 0.8AB | 175.0 ± 5.0A | 24.2 ± 1.3B | 83 ± 3C | 6.0 ± 0.4B | 18.0 ± 0.8C | 3.01 ± 0.15BC | 0.075 ± 0.005AB | 0.13 ± 0.02B | 21 ± 3AB | 22 ± 2A |
| RS_H2 | 5.31 ± 0.21BC | 37.2 ± 2.0CD | 1.49 ± 0.16AB | 0.35 ± 0.05AB | 16.5 ± 0.7AB | 170.2 ± 4.9A | 28.9 ± 0.8A | 142 ± 6B | 5.7 ± 0.5B | 23.7 ± 0.7A | 2.92 ± 0.11C | 0.064 ± 0.008B | 0.17 ± 0.02A | 24 ± 3A | 26 ± 1A |
| FS_1 | 5.01 ± 0.15C | 42.5 ± 2.0ABC | 1.70 ± 0.04AB | 0.40 ± 0.03A | 17.3 ± 0.9A | 143.8 ± 3.5B | 12.9 ± 0.6C | 198 ± 8A | 4.1 ± 0.6C | 22.0 ± 3.0AB | 3.35 ± 0.25B | 0.061 ± 0.008B | 0.19 ± 0.03A | 15 ± 2BC | 25 ± 3A |
| FS_2 | 5.47 ± 0.27ABC | 47.5 ± 3.2A | 1.83 ± 0.03A | 0.26 ± 0.04AB | 18.2 ± 0.2A | 103.5 ± 3.3C | 7.6 ± 0.4D | 45 ± 3E | 3.5 ± 0.3C | 15.5 ± 1.2C | 4.07 ± 0.07A | 0.064 ± 0.007B | 0.07 ± 0.01D | 16 ± 2BC | 23 ± 3A |
| RS_D | 4.49 ± 0.08D | 33.8 ± 0.7D | 1.41 ± 0.06B | 0.33 ± 0.05AB | 16.6 ± 0.6AB | 98.3 ± 2.1C | 24.3 ± 0.9B | 96 ± 6C | 9.7 ± 0.4A | 16.6 ± 0.5C | 2.77 ± 0.14C | 0.100 ± 0.020A | 0.09 ± 0.01CD | 18 ± 2ABC | 21 ± 4A |
| Mean | 5.28 | 41.1 | 1.63 | 0.31 | 16.8 | 142.3 | 16.8 | 105 | 5.5 | 19.1 | 3.16 | 0.075 | 0.13 | 18 | 23 |

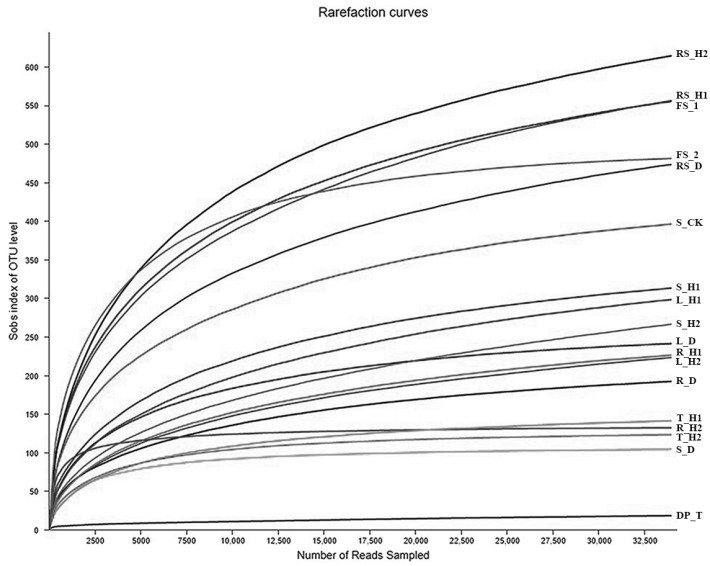

**Figure 1** **Rarefaction curves of fungal communities based on observed operational taxonomic units (OTUs) for 18 plant and soil samples under an *Atractylodes macrocephala* continuous cropping system.** Each plant and soil samples (18 samples) were randomly divided into three subsamples (18 × 3), and were analyzed using the average data of their subsamples, respectively. One-year fallow soil (FS_1), 2-year fallow soil (FS_2), 1-year healthy rhizosphere soil (RS_H1), 2-year healthy rhizospheric soil (RS_H2), 2-year root-rot diseased rhizospheric soil (RS_D), blank control soil (no *A. macrocephala* was planted) (S_CK), 1-year-old healthy root/stem/leaf/tuber (R/S/L/T_H1), 2-year-old healthy root/stem/leaf/tuber (R/S/L/T_H2), 2-year-old root-rot diseased root/stem/leaf/tuber (R/S/L/T_D).

## Composition and alpha diversity of the fungal communities

A total of 484,844 and 251,911 reads were obtained from *A. macrocephala* plant and soil samples, respectively. A total of 48,550 reads mapped to unidentified phylum, while 271,914 reads mapped to unidentified genera. As indicated by the rarefaction curves, all samples reached the saturation phase with a satisfactory level of confidence and a Good's coverage index ≥ 99.59% (Table 1, Fig. 1). Sequences from plant samples clustered into 2,545 OTUs using a 3% dissimilatory threshold. The core fungi of plant samples consisted of 276 OTUs, each of which had a relative abundance of greater than 1% and were found in 95% of plant samples, and included Pleosporales (with 73 OTUs), Hypocreales (45), Capnodiales (81), Helotiales (33), Tremellales (19), Cantharellales (5) and Sporidiobolales (20) (Fig. 2A). Samples T_D and T_H2 contained the highest levels of Hypocreales (93.43%) and Pleosporales (72.57%) sequences, respectively (Fig. 2C).

Sequences from soil samples clustered into 3,596 OTUs, and their core fungal composition (431 OTUs) included Eurotiales (63), Hypocreales (104), Pleosporales (65), Chaetothyriales (51), Mortierellales (27), Sordariales (44), Helotiales (42), Geminibasidiales (3), Chaetosphaeriales (12) and Capnodiales (20) (Fig. 2B). Samples S_CK and FS_1 contained the highest levels of Eurotiales (40.08%) and Hypocreales (37.04%), respectively (Fig. 2C). At the phylum level, Zygomycota, Ascomycota, and Basidiomycota were the dominant fungal phyla in both continuous cropping and fallow samples. During continuous

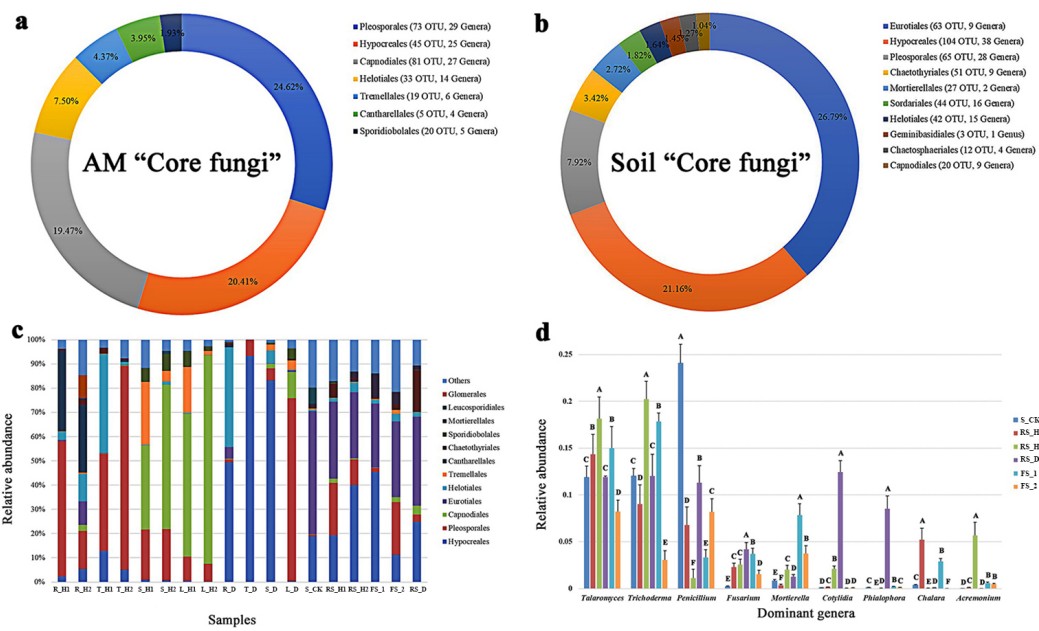

**Figure 2** **Fungal composition of 18 plant and soil samples under an *A. macrocephala* (AM) continuous cropping system.** Each plant and soil samples (18 samples) were randomly divided into three subsamples (18 × 3), and were analyzed using the average data of their subsamples, respectively. The "core fungi" of *A. macrocephala* (A) and soil (B). The relative abundance at order level of 18 plant and soil samples (C), and the dominant genera of six soil samples (D), different letters above the bars in one genus denote significant differences at $P < 0.01$. One-year fallow soil (FS_1), 2-year fallow soil (FS_2), 1-year healthy rhizosphere soil (RS_H1), 2-year healthy rhizospheric soil (RS_H2), 2-year root-rot diseased rhizospheric soil (RS_D), blank control soil (no *A. macrocephala* was planted) (S_CK), 1-year-old healthy root/stem/leaf/tuber (R/S/L/T_H1), 2-year-old healthy root/stem/leaf/tuber (R/S/L/T_H2), 2-year-old root-rot diseased root/stem/leaf/tuber (R/S/L/T_D).

cropping, rhizospheric soil showed a significant increase of *Talaromyces*, *Fusarium*, *Cotylidia* and *Acremonium*, and significant decrease of *Penicillium* ($P < 0.01$) (Fig. 2D). A higher abundance of the genera *Cotylidia*, *Phialophora*, and *Fusarium* in rhizospheric fungi was observed in root-rot diseased plants than the S_CK (Fig. 2D). Additionally, rhizospheric soil fungal alpha-diversity decreased with long-term *A. macrocephala* cropping. The fungal diversity index was the highest in blank control soil, and the fungal diversity of the fallow soil samples was also higher than that in samples from continuous cropping. Moreover, root-rot disease decreased the endophytic fungal diversity of *A. macrocephala* (Table 1).

## Structure of the fungal communities

We performed a hierarchical clustering analysis on the top 25 most abundant fungal genera across 18 plant and soil samples (Fig. 3A). The analysis showed that *A. macrocephala* aerial samples clustered into one category (except for T_D), while all rhizospheric and fallow soil samples clustered into another category. R_D, T_H1, R_H1, and T_H2 endophytic samples clustered together, but were separate from the R_H2 sample. PCA analysis based on the OTU composition revealed obvious variations in fungal communities among the 18 plant and soil samples (Fig. 3B). The first two axes (PC1 and PC2) explained 41.59% and

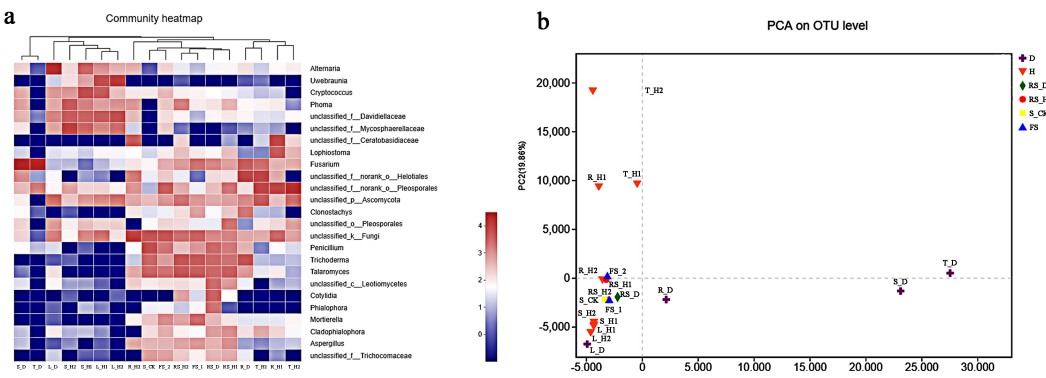

**Figure 3** **The top 25 most abundant fungal communities across 18 plant and soil samples (A) and principle component analysis (PCA) based on the operational taxonomic unit (OTU) composition (B).** Each plant and soil samples (18 samples) were randomly divided into three subsamples (18 × 3), and were analyzed using the average data of their subsamples, respectively. One-year fallow soil (FS_1), 2-year fallow soil (FS_2), 1-year healthy rhizosphere soil (RS_H1), 2-year healthy rhizospheric soil (RS_H2), 2-year root-rot diseased rhizospheric soil (RS_D), blank control soil (no *A. macrocephala* was planted) (S_CK), 1-year-old healthy root/stem/leaf/tuber (R/S/L/T_H1), 2-year-old healthy root/stem/leaf/tuber (R/S/L/T_H2), 2-year-old root-rot diseased root/stem/leaf/tuber (R/S/L/T_D).

19.86% of the total variance in the fungal OTUs of the plant and soil samples, respectively, in *A. macrocephala* continuous cropping fields.

## Correlation between soil properties and the fungal communities

We then removed the redundant variables, and the eight remaining environmental characteristics were subjected to detrended correspondence analysis (DCA). The DCA showed that the responses of fungal community composition to soil properties fit a single-peak model (Length = 6.32). Therefore, we further analyzed how fungal communities associated with soil physicochemical factors based on CCA (Fig. 4). Soil physicochemical factors Hg, electrical conductivity, available P, Cr, pH, and hydrolysis N positively correlated with each other, while total N and organic matter positively correlated with each other. Moreover, pH ($p = 0.001$), hydrolysis N ($p = 0.001$), electrical conductivity ($p = 0.001$), and Hg ($p = 0.001$) had a significant correlation with fungal community structure (Fig. 4). Spearman's correlation analysis revealed that the relative abundance of the genus *Acremonium* was significantly negatively corelated with soil electrical conductivity and Hg content, while *Penicillifer* genus abundance had a negative correlation with soil Cr content (Tables S1, S2).

## DISCUSSION

Sustainable *A. macrocephala* production management calls for a deep understanding of how continuous cropping alters the structure and diversity of fungal communities. To gain more insight in this, we assessed the effects of continuous cropping on fungal communities in rhizospheric soil, rotation soil and endophytes of *A. macrocephala*. We found that the diversity, structure and composition of rhizospheric soil and endophyte fungal communities were greatly affected by continuous cropping. Rhizospheric soil fungal

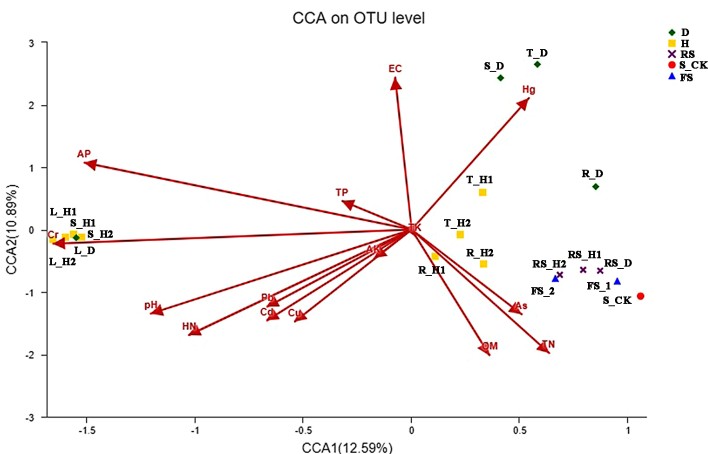

**Figure 4  Canonical correspondence analysis (CCA) between the soil and fungal community.** Each plant and soil samples (18 samples) were randomly divided into three subsamples (18 × 3), and were analyzed using the average data of their subsamples, respectively. One-year fallow soil (FS_1), 2-year fallow soil (FS_2), 1-year healthy rhizosphere soil (RS_H1), 2-year healthy rhizospheric soil (RS_H2), 2-year root-rot diseased rhizospheric soil (RS_D), blank control soil (no *A. macrocephala* was planted) (S_CK), 1-year-old healthy root/stem/leaf/tuber (R/S/L/T_H1), 2-year-old healthy root/stem/leaf/tuber (R/S/L/T_H2), 2-year-old root-rot diseased root/stem/leaf/tuber (R/S/L/T_D).

alpha-diversity indices, including Chao1 and Shannon indices, decreased after continuous cropping practices. In addition, fallow soils showed relatively higher overall fungal activity, whereas fungal diversity was similar between continuously cropped soil and the S_CK, suggesting that fungal diversity gradually recovers to unplanted control soil with time of fallowing. Severe root-rot disease has long been known as a major problem in continuous cropping (*Tan et al., 2017a*). Consistently, we detected a relatively higher fungal community diversity in healthy tissues and rhizospheric soils compared to diseased samples. Therefore, the decrease of soil and endophytic fungal diversity may play a role in disease development during *A. macrocephala* continuous cropping.

Rhizospheric soil fungal communities were also strongly influenced by continuous cropping of *A. macrocephala*. Zygomycota, Ascomycota, and Basidiomycota were the dominant fungal phyla in both continuous cropping and fallow samples. At the genus level, the relative abundance of *Talaromyces* and *Fusarium* increased significantly with cropping time. *Talaromyces* and *Fusarium* genera fungi contain several potential pathogens, such as *Fusarium oxysporum* and *Talaromyces helices* (*Wu et al., 2016*). Thus, increases in these genera may negatively contribute to *A. macrocephala* continuous cropping, eventually leading to increased disease pressure. Moreover, the fungal community profile of fallow soil became more similar to the control soil profile as the number of years of fallowing increased, indicating that recovery of a healthy soil profile is possible.

Endophytic fungi usually inhabit different plant tissues without harming their hosts. However, we observed reduced abundance of endophytic fungal OTUs in diseased samples, signifying reduced endophytic fungal diversity. Ascomycota, Basidiomycota, and Zygomycota were the top three fungal phyla detected in both healthy and diseased

*A. macrocephala* plants, which was similar to the profile of soil fungi. This pattern fits with earlier publications, which have shown that endophytic fungi are primarily derived from soil fungi that enter the plants via roots, tubers, leaves and stems (*Dai et al., 2010*; *Tan et al., 2017b*). At the genus level, *Fusarium* and *Alternaria* abundance significantly increased in the root-rot diseased *A. macrocephala* samples. This may be due to increases in other root-rot disease associated species, such as *F. oxysporum*, *F. solani*, and *Alternaria gansuense*. Increases in these pathogenic fungi likely caused a decrease in endophytic fungi due to limited availability of space and nutrients (*Zeng, 2016*; *Zhang et al., 2015*). These findings warrant future studies on the changes in microbial communities, as they can be used as indicators of overall soil and plant health during continuous cropping.

A deeper understanding of soil properties during *A. macrocephala* continuous cropping systems is key to improving soil productivity. Long-term monoculture of *A. macrocephala* has been reported to reduce organic matter content and soil pH, due to the return of organic material and the application of fertilizer to the soil (*Geng et al., 2015*). Increased soil hydrolysis N, available P, and available K contents often result from fertilizer application (*Shi, 2018*). The optimum pH range for *A. macrocephala* growth is 5.1 to 6.6 (*Zhu et al., 2018b*), and the pH decline seen during continuous cropping may therefore increase the disease susceptibility of *A. macrocephala*.

In agricultural ecosystems, soil microbial communities have a major impact on soil organic matter accumulation and nutrient cycling, which are often used as indicators of soil quality (*Ashworth et al., 2017*). In our study, soil pH, hydrolysis N, electrical conductivity, and Hg content were most strongly correlated with fungal community structure during continuous cropping of *A. macrocephala*. Soil pH may directly alter fungal community composition by inhibiting fungal survival and growth, as seen in earlier publications indicating fungal taxa were unable to grow under a certain soil pH (*Zhang et al., 2016*). In addition, soil communities can affect soil N dynamics, while hydrolysis N has a major impact on the composition of fungal community under continuous cropping. The change in fungal composition may be a result of both fertilization and the observed increase in hydrolysis N cycling (*Thompson & Kao-Kniffin, 2019*). Agricultural management practices, such as high fertilizer usage and restricted irrigation can immensely influence electrical conductivity variation (*Adviento-Borbe et al., 2006*; *Kim et al., 2016*). Electrical conductivity is associated with soil salinity and our results suggest it may be an important predictor of fungal community compositions in continuous cropping soils of *A. macrocephala*. Moreover, *Acremonium* and *Penicillifer* show a negative correlation with heavy metal contents in the soil, especially Hg and Cr.

## CONCLUSIONS

Overall, we found that continuous cropping and severe root-rot disease could both significantly affect the structure and diversity of *A. macrocephala* endophytic and soil fungal communities. Rhizospheric soil pH and organic matter content decreased with increasing continuous cropping time. Moreover, the abundance and diversity of fungal communities decreased, while the prevalence of severe root-rot disease increased with

prolonged continuous cropping. Of all root-rot rhizospheric and endogenous fungal species, *Fusarium* was most significantly enriched upon continuous cropping. Further, soil pH, hydrolysis N, electrical conductivity, and Hg were most strongly correlated with fungal community composition. Simultaneously examining both endophytic fungal populations (from surface-sterilized plant tissues) and the rhizospheric samples (including soils from the roots) allowed us to gain a deeper understanding of how continuous cropping alters fungal populations. Our results suggest that changes in fungal diversity can be used to predict disease outbreaks in *A. macrocephala* continuous cropping systems. The findings of this research can also guide the development of management strategies to improve *A. macrocephala* production.

### Funding
This study was supported by the National Natural Science Foundation of China (81673528), and the Opening Project of Zhejiang Provincial First-rate Subject (Chinese Traditional Medicine), Zhejiang Chinese Medical University (Ya2017001). The funders had no role in study design, data collection and analysis, decision to publish, or preparation of the manuscript.

### Grant Disclosures
The following grant information was disclosed by the authors:
National Natural Science Foundation of China: 81673528.
Opening Project of Zhejiang Provincial First-rate Subject (Chinese Traditional Medicine).
Zhejiang Chinese Medical University: Ya2017001.

### Competing Interests
The authors declare there are no competing interests.

### Author Contributions
- Bo Zhu conceived and designed the experiments, performed the experiments, analyzed the data, prepared figures and/or tables, authored or reviewed drafts of the paper, and approved the final draft.
- Jianjun Wu and Qingyong Ji performed the experiments, prepared figures and/or tables, and approved the final draft.
- Wei Wu, Shihui Dong and Jiayan Yu analyzed the data, prepared figures and/or tables, and approved the final draft.
- Qiaoyan Zhang conceived and designed the experiments, prepared figures and/or tables, authored or reviewed drafts of the paper, and approved the final draft.
- Luping Qin conceived and designed the experiments, authored or reviewed drafts of the paper, and approved the final draft.

### Data Availability
The data are available at NCBI at SRA (SRP238130) and BioProject (PRJNA596555).

## Supplemental Information

Supplemental information for this article can be found online at http://dx.doi.org/10.7717/peerj.8905#supplemental-information.

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
