# Peer review of "Diversity of rhizosphere and endophytic fungi in Atractylodes macrocephala during continuous cropping"

_PeerJ, doi:10.7717/peerj.8905_

## Round 0.1 · original submission · Major Revisions

Please find the comments by two independent reviewers. Although both reviews are rather short, they both identify key deficiencies in your manuscript. Re-submit a version that incorporates the suggestions proposed by the reviewers. Pay special attention to the English grammar and structure and make sure to submit your primary Next Generation Sequencing data to NCBI-SRA, providing a corresponding SRA ID in the manuscript. Also make sure to provide enough data as to how the experiments (i.e., sequencing experiments) were performed.

Reviewer 1 ·

Basic reporting

The authors used culture-independent Illumina MiSeq to evaluate the endophytic and rhizospheric fungi upon continuous cropping of A. macrocephala. The objectives of the paper are clearly stated in the introduction. The introduction is well structured and referenced as is the discussion. The authors shared the raw data and I suggest indicating in the manuscript the deposit of the data in some bank.

Experimental design

The material and methods are unclear about the experimental design. The authors need to clearly indicate how many biological replicas and how many technical replicas have been sequenced. Apparently, the authors pooled samples. Statistical analyzes are committed to this approach. Other comments are in the general comments.

Validity of the findings

I indicated my comments on data validation in the item “General comments for the author”

Additional comments

Line 47: term “microflora” is not appropriate. I suggest replacing the term microbiota.

Line 53: Authors could insert recent references to the first sentence.

M&M:
The caption of fig 1 is confusing. Consider the abbreviations used in the material and methods: 1-year fallow soil (S_F1), 2-year fallow soil (S_F2), 1-year healthy rhizospheric soil (RS_H1), 2-year healthy rhizospheric soil (RS_H2), 2-year root-rot diseased rhizospheric soil (RS_D), blank control soil with no A. macrocephala planted (S_CK)

Improve the text so that it is clear: five zones? How many plots? Replicates?
How deep was the soil sampled? Composite sampling (how many subsamples?)?
How long were samples processed after storage?

Line 116: Were the samples sequenced separately or pooled for sequencing? Make it clear how many biological replicas and how many technical replicas were used.
Line 117: The authors used which parameters to filter the reads: size, phred value, etc:
Were chimeras excluded? Have mitochondrial sequences been deleted? Communities have different amounts of reads. Did the authors rarefy and normalize data before analysis? Rarefaction is essential for comparing libraries with different amounts of reads.
How many reads have not been identified?
Did the authors adjust the significance value in multiple comparisons, for example using false discovery rate (FDR)?

Results:
The material and methods need to be clear in the sample design. The variation indicated in fig2d came from how many sample units? Did these sample units come from sample pools?
Fig2d: Which parameter to select a dominant genera?
Line 157: Indicate the statistical test together with the value of p.

Line 160: “The fungal diversity index was the highest in blank control soil, and the fungal diversity of the fallow soil samples were also higher compared with samples from continuous cropping”: How to state in larger or smaller values without statistical analysis?

Line 166: What criteria for choosing 25 most abundant fungi?

Discussion:
The discussion needs to clearly indicate whether or not there was a limitation in the sample design.

Reviewer 2 ·

Basic reporting

The overall English language needs to be improved for clarification.

More references for background introduction are needed.

Experimental methods and data presentations need to provide more details and support for the conclusions.

Experimental design

The experimental design is OK. However, the method needs to provide more details, for instance:

1. The soil sample collection, how to collect rhizosphere vs. endosphere samples; and how the Furasrium involved in the related study, any related experiments and the plants showed any related symptom?

2. How the sequencing library was set up and how the related bar codes were used?

3. All the soil content analysis did not show the standard deviation. A more detailed statistical analysis needs to be provided for clarification.

Validity of the findings

The methods, data presentation, and statistical analysis need to be improved and clarified to draw more accurate conclusions.

---

## Round 0.2 · Minor Revisions

I requested your manuscript to be re-visted by the original reviewers. Unfortunately, only one of the three invited reviewers responded to the request. The review we received points out remaining deficiencies with your manuscript. Please attend to these requests and re-submit your work as soon as possible. I am looking forward at receiving your revised manuscript.

Reviewer 3 ·

Basic reporting

The authors used culture-independent Illumina MiSeq to evaluate the endophytic and rhizospheric fungi upon continuous cropping of A. macrocephala. The objectives of the paper are clearly stated in the introduction. The introduction is well structured and referenced as is the discussion. The authors shared the raw data and I suggest indicating in the manuscript the deposit of the data in a gene-bank.

Experimental design

- If I understand correctly, there were 54 samples totally. However, Figure 2-3 and Table 1 indicates there were 18 samples, I am very confused about this point,although I read you explanation in the responds letter.

- Is it possible used abbreviation for the samples like this?
1-year fallow soil (FS_1),
2-year fallow soil (FS_2),
1-year healthy rhizospheric soil (RS_H1),
2-year healthy rhizospheric soil (RS_H2),
2-year root-rot diseased rhizospheric soil (RS_D)
1-year-old healthy root/stem/leaf/tuber (R/S/L/T_H1),
2-year-old healthy root/stem/leaf/tuber (R/S/L/T_H2),
2-year-old root-rot diseased root/stem/leaf/tuber (R/S/L/T _D)

- Bioinformatics analysis is not part of statistical analysis, usually. (see details in the file attached)

- More details of the statistical analysis could be added. (see details in the file attached)

- the endophytic fungi determined by plants much more than soil. So, what is the point to analysis correlation between endophytic fungi and soil?

Validity of the findings

It is very import to described how statistical analysis conducted and show the results clearly. Because in this article it is not difficult to evaluate the validity of the findings without statistical analysis.

Additional comments

I indicated the comments in the file attached.

Annotated reviews are not available for download in order to protect the identity of reviewers who chose to remain anonymous.

---

## Round 0.3 · accepted · Accept

I apologize for the delay in the review process. I have read your rebuttal letter and I think that your work is now ready for publication. Thanks for your patience.